# Secondary Narcolepsy as Worsening Sign in a Pediatric Case of Optic Pathway Glioma

**DOI:** 10.3390/children9101455

**Published:** 2022-09-23

**Authors:** Beatrice Laus, Anna Maria Caroleo, Giovanna Stefania Colafati, Andrea Carai, Romina Moavero, Michela Ada Noris Ferilli, Massimiliano Valeriani, Angela Mastronuzzi, Antonella Cacchione

**Affiliations:** 1Hispalense Institute of Pediatrics, 41014 Seville, Spain; 2Department of Onco-Hematology, Cell Therapy, Gene Therapy, and Hemopoietic Transplant, Bambino Gesù Children’s Hospital (IRCCS), Piazza Sant’Onofrio 4, 00165 Rome, Italy; 3Department of Imaging, Bambino Gesù Children’s Hospital (IRCCS), Piazza Sant’Onofrio 4, 00165 Rome, Italy; 4Neurosurgery Unit, Department of Neurosciences, Bambino Gesù Children’s Hospital (IRCCS), 00165 Rome, Italy; 5Child Neurology and Psychiatry Unit, University Tor Vergata, 00133 Rome, Italy; 6Neurology Unit, Department of Neurosciences, Bambino Gesù Children’s Hospital (IRCCS), Piazza Sant’Onofrio 4, 00165 Rome, Italy; 7Center for Sensory-Motor Interaction, Aalborg University, 9100 Aalborg, Denmark

**Keywords:** brain tumor, low-grade gliomas, optic pathway glioma, hypersomnia, pediatric case report, neurological sequelae

## Abstract

Narcolepsy, a neurologic disorder that leads to excessive daytime sleepiness, may represent a rare consequence of neoplastic lesions involving the sellar/parasellar and hypothalamic regions, the anatomical areas responsible for wakefulness. Optic pathway gliomas represent the most common neoplasm of these regions and present an excellent overall survival, while long-term neurologic impairments, such as visual loss, endocrinopathies, or sleep disorders, are the principal causes of morbidity. In this case report, we describe a non-NF1 patient suffering from a very extensive optical pathway glioma, who several years after the diagnosis in a radiological condition of stable disease, presented with severe narcolepsy, a rare complication, that led to the death of the patient.

## 1. Introduction

Narcolepsy is a neurologic disorder characterized by excessive daytime sleepiness and sudden sleep attacks, associated with cataplexy and rapid eye movement (REM) sleep-related phenomena. It is a rare pediatric condition that is predominantly idiopathic, and in its secondary form can sometimes be associated with central nervous system (CNS) tumors. This occurrence is very rare, although the exact prevalence of excessive daytime sleepiness in pediatric neuro-oncology patients is difficult to assess; several case series of secondary narcolepsy have been published in adults, whereas fewer notions are available in children [1,2,3,4,5,6,7,8]. Orexin deficiency is the major cause of this disorder and, in the case of CNS tumors, is potentially attributable to the direct damage (caused by surgery, radiation, or tumor infiltration itself) of the hypothalamic neurons that produce this excitatory neurotransmitter [9]. The treatment of narcolepsy is based on the use of drug therapy targeting dopaminergic and noradrenergic or histaminergic pathways [10]. These therapies are generally well tolerated and lead to symptom improvement.

Tumors in the hypothalamic/pituitary region have been linked to the development of narcolepsy in childhood, and craniopharyngiomas, pituitary adenomas, and low-grade gliomas (LGG) are among the most common [9,11,12].

Optic pathway gliomas (OPGs) represent the LGG that can develop anywhere in the optic pathway, including one or both optic nerves, chiasm, or hypothalamus, but can extend to and infiltrate the adjacent brain structures [13,14,15,16]. OPG presents an excellent prognosis, with overall survival (OS) that can reach up to 100% in those located exclusively in the optic nerve. However, when the location is very close to the hypothalamus and pituitary gland, the surgical treatment becomes challenging, and the tumor and its treatment can lead to long-term neurologic impairments, such as visual loss and endocrinopathies or sleep disorders [17,18,19,20,21].

We describe the case of a 3-year-old patient affected by OPG and never treated with radiotherapy, who presented with a sudden state of drowsiness that evolved into narcolepsy. The rapid clinical deterioration occurred several years after the diagnosis and in a phase of radiological stability of the tumor

## 2. Case Presentation

A 3-year-old boy came to our attention for visual impairment, convergent strabismus of the left eye, and evidence of hypotrophy of the optic disc in the fundus. He had a history of seizures with onset at 18 months treated with sodium valproate.

At physical examination, the patient was in adequate general condition and neurocognitive development, severe bilateral hypovision (bilateral optical hypotrophy, with OCT (optical coherence tomography) highlighting a retinal nerve fiber layer thickness (RNFL) severe reduction, visual loss quantifiable with Lea figures in 2/10 in left eye and 4/10 in right eye) horizontal nystagmus, divergent strabismus of the left eye and poor pupillary reaction to photostimulation.

A single *café au lait* spot was detected, but neurofibromatosis type 1 (NF1) was previously excluded by genetic analysis.

A magnetic resonance imaging (MRI) of the brain and spinal cord documented the presence of an infiltrative, space-occupying lesion involving the optic-chiasmatic region (Figure 1).

The patient underwent a biopsy (stereotactic robot-assisted needle biopsy by a right transfrontal route of the lesion) that allowed the diagnosis of ganglioglioma grade 1 in accordance with the latest edition of the World Health Organization (WHO) classification of central nervous system (immunohistochemistry: positivity for CD34+, GFAP and OLIG2; p53 negative, BRAFV600 negative both in immunohistochemistry and molecular analysis).

Chemotherapy treatment was started according to the International Society of Paediatric Oncology-Low Grade Glioma (SIOP LGG) protocol [22].

Almost one year after starting therapy, the patient began to show a clinical picture of cachexia, and a brain MRI showed disease progression (Figure 2). A new ophthalmological evaluation confirmed a further loss of visual acuity with complete bilateral hypovision. Therefore, a second-line chemotherapy treatment as first and target therapy as secondary were started. Five years later (at 8 years of age) the patient presented to the emergency department drowsy but arousable by verbal or tactile stimuli and well oriented when awake. A brain computed tomography (CT) scan and a new MRI were performed, showing a stable disease concerning the most recent MRI.

The possibility of hydrocephalus contributing to clinical deterioration was considered but MRI did not show any signs of raised intracranial pressure. Other systemic causes of somnolence such as electrolyte imbalance or anemia were excluded.

The possibility of a surgical procedure was discussed at the tumor board and considered futile since the tumor had an infiltrative growth and imaging did not show any signs of intracranial hypertension.

The excessive daytime sleepiness (EDS) associated with sleep paralysis and hypnopompic hallucinations (upon awakening) did raise the suspicion of secondary narcolepsy in differential diagnosis with autoimmune encephalitis.

Autoimmune antibodies (Anti GAD, Anti Hu, Anti Yu, Anti NMDA, and voltage-gated K channel antibody) were negative. A multi-sleep latency test (MSLT) performed after a video-polysomnography recording more than 6 h of nighttime sleep demonstrated the presence of excessive daytime sleepiness and a very short sleep onset latency (<2 min) with 2 SOREM (sleep onset REM), thus resembling a narcolepsy pattern. Treatment for narcolepsy was started with modafinil, which was only partially effective in decreasing daily somnolence and caused side effects such as tachycardia and psychomotor agitation.

Pitolisant as an add-on was then tried, but it was associated with severe anxiety symptoms without benefit on the hypersomnolence; therefore, it was withdrawn.

In the following months, the child developed a progressive and significant neurological decline with the appearance of continuous dyskinesias, psychosis, mutism, encopresis, enuresis, and secondary autism spectrum disorders. The neurological impairment led to the death of the child within a few months.

## 3. Discussion

Narcolepsy is defined as a debilitating condition of excessive daytime sleepiness and sudden attacks of sleep and can be accompanied by cataplexy, sleep paralysis, hallucinations, and disrupted nocturnal sleep [8,12,23,24,25,26,27]. However, in younger patients, narcolepsy can be manifested with hyperactivity, aggression, and irritability [11]. The prevalence of narcolepsy in childhood is estimated to be approximately 0.02–0.05% [12]. In most cases, narcolepsy is a primary pathological condition that results in the loss of hypocretin neurons in the posterolateral hypothalamus. Recently, an autoimmune process targeting these neurons has been hypothesized, due to the identification of some autoantibodies (Tribbles homolog2-specific antibody) in serum and cerebrospinal fluid (CSF) of narcolepsy patients [12,28]. In this broad spectrum of autoimmune pathogenesis, several autoantibodies have been checked in blood samples of narcoleptic patients, but none has been detected in a consistent manner. For this reason, narcolepsy still does not fully meet the criteria for being classified as a genuine autoimmune disease [28].

However, indirect evidence of an autoimmune pathogenesis of narcolepsy is provided by the observation that narcolepsy occasionally occurs in association with paraneoplastic syndromes and other autoimmune diseases, such as multiple sclerosis, coeliac disease and systemic lupus erythematosus [29]. In our patient, autoantibodies Anti GAD, Anti Hu, Anti Yu, Anti NMDA, and voltage-gated K channel antibody, exploring an autoimmune pathogenesis of somnolence resulted negative.

A genetic predisposition has also been described involving a specific HLA allele (HLA-DQB1*0602) found in 85–95% of patients with narcolepsy [24].

Narcolepsy may rarely result from secondary causes such as inherited disorders, encephalitis, head trauma, stroke, vascular malformation, or CNS tumors [30]. Patients with tumors of the sellar/parasellar and hypothalamic region are the most vulnerable to develop these types of disorders, as orexin producing cells are located exclusively in the hypothalamus. The exact prevalence of excessive daytime sleepiness in children with cancer is difficult to assess; furthermore, sleep complaints reported by this population of children with CNS tumors additionally include fatigue (often not differentiated from sleepiness and more difficult to quantify), respiratory insufficiency (in children with tumors involving the brainstem), hypoxia, insomnia, circadian rhythm disorder, nocturnal seizures and snoring [1,2,3,4,5,6,7].

Mechanisms by which cancer disrupts circadian rhythm are still unclear. One mechanism hypothesized to achieve this is via DNA methylation of specific clock genes, impacting both the peripheral circadian system and the central circadian clock, potentially leading to sleep problems. [3]

The regulation of wakefulness, sleep, and circadian rhythm is preserved through neuropeptide-producing neuronal networks involving the thalamus, hypothalamus, basal forebrain, and brain stem. When the tumor is located in these areas due to its position and/or its expansion as well as to surgery or cranial irradiation, the sleep-wake (SW) regulatory systems can be affected, disturbing the signaling pathways [31,32].

Following the appearance of characteristic symptoms, which include excessive daytime sleepiness, sudden attacks of sleep or cataplexy, the suspicion of primary or secondary narcolepsy is based on the performance of nocturnal polysomnography (PSG) and the multiple sleep latency tests (MSLT), a test that measures the tendency and rapidity of falling asleep under standardized conditions [9,11,33,34]. It is also possible to assay hypocretin/orexin levels in cerebrospinal fluid, which are usually low (<110 pg/mL) in patients with lesions affecting the hypothalamus [24,25,30,35,36].

The treatment of narcolepsy is the same regardless of its etiopathogenesis. It is based on the use of drug therapy targeting dopaminergic (modafinil, armodafinil), dopaminergic, and noradrenergic (methylphenidate, solriamfetol, or dextroamphetamine) or histaminergic (pitolisant) pathways [10]. Another possible treatment is with sodium oxybate, a CNS depressant with an agonist action on GABA_B_ receptors [37]. These therapies are generally well tolerated and lead to symptoms improvement. Those that are most frequently associated with narcolepsy tumors in childhood involve the hypothalamus, brainstem, pituitary gland, and thalamus, and appear to be craniopharyngiomas, germinomas, adenomas, and gliomas [3,9,11]. According to the literature, tumor location and consequently the damaged area, appears to be more important than the specific lesion type itself in the development of sleep disorders [35,38]. Associated risk factors seem to be obesity and received radiation dose > 30 Gy [9,11].

Pediatric series of narcolepsy in pediatric patients with brain tumors are few and the percentage of incidence of this neurological complication is different among the available series. This evidence is due to the difficulty in defining the exact diagnosis of this sleep disorder.

In Table 1, we described the most significant literature to date. Pickering et al. analyzed a population of 61 children with a diagnosis of a primary brain or spinal tumor. They divided the patients into two groups according to the location of the tumor. Patients fulfilled questionnaire surveys to assess sleep quality, fatigue, and mental health status, and polysomnography and multiple sleep latency tests were performed. They observed narcolepsy in 8% of patients, and they found that patients with tumors involving the sleep-wake regulatory areas were sleepier/more fatigued [32].

In another study, Mandrell et al. analyzed a population of 110 pediatric patients with the diagnosis of a craniopharyngioma; the patients completed a baseline sleep clinic evaluation performing a PSG and MSLT. Among patients who completed PSG and MSLT, a diagnosis of excessive daytime sleepiness was found in 80% of them.

Particularly, hypersomnia was diagnosed in 45% and narcolepsy in 35%, with a higher percentage in overweight or obese patient groups [34].

Craniopharyngiomas are slow-growing tumors of the sellar and parasellar region and may also involve the hypothalamus; they are the most common intracranial tumors of nonglial origin in the pediatric population, representing 6–9% of pediatric brain tumors [11,39,40,41]. Patients affected by craniopharyngioma involving the hypothalamic region, or with hypothalamic injury occurring during resection, are the most vulnerable to sleep disorders including excessive daytime [39,42].

One of the hypothesized mechanisms leading to daytime sleepiness in craniopharyngioma patients is the decreased melatonin production, documented in patients with craniopharyngioma, and caused by the involvement of the hypothalamic suprachiasmatic nucleus (SCN), a key component of the sleep-wake regulatory system and regulation of circadian rhythm. Indeed, it directs the diurnal secretion of melatonin from the pineal gland in response to the environmental light/dark cycle, and the melatonin itself is involved in sleep promotion and regulation of circadian rhythms [40,43]. Moreover, Mandrell et al. highlighted that craniopharyngioma patients with severe obesity (BMI > 4 sd) presented more severe daytime sleepiness than normal weight or less obese craniopharyngioma patients. Several studies have shown how obese craniopharyngioma patients present lower salivary melatonin concentrations at night compared to other groups, possibly due to the correlation between BMI and the degree of hypothalamic injury, as some authors hypothesize) [31,35,36,43,44,45].

Weil et al. described the case of an 11-year-old girl diagnosed with a large sellar and suprasellar germinoma, in which narcolepsy led to the diagnosis and thus subsequently disappeared during treatments [30]. In a further study, Rosen et al. analyzed a population of 14 children with CNS tumors and sleep disorders by performing the MSLT: nine complained of excessive daytime sleepiness (EDS) and five met the criteria for narcolepsy [35]. One patient presented a hypothalamic Langerhans cell histiocytosis, two patients had a suprasellar tumor (a craniopharyngioma and an astrocytoma) and two did not have tumors involving the hypothalamus (both were posterior fossa tumors) but were subjected to neurosurgical procedure and received chemotherapy and cranial radiation. Four of these patients received treatment with stimulants, and all treated patients responded favorably to therapy. In a subsequent study, the same authors analyzed a population of 70 pediatric patients with cancer and sleep problems: 48 patients (68%) were affected by a CNS tumor. They confirmed that the hypothalamic, thalamic, and brainstem tumor location seemed to be more relevant than tumor type as a risk factor [38].

Tachibana et al. presented a case of an 11-year-old girl who developed narcolepsy following surgical removal of a craniopharyngioma with almost total removal of the hypothalamus that was seen on MRI, confirming the aforementioned hypothesis. Beyond the craniopharyngioma, LGGs are the most common CNS tumors in childhood found in the hypothalamic/chiasmatic region; LGGs account for approximately one-third of all SNC tumors [46,47,48,49,50,51]. The 10-year overall survival rate is between 85–96% [48]. However, patients affected by pLGGs can develop functional, neurologic, and endocrine complications, deriving from the disease and/or treatment [52].

OPGs account for approximately 3–5% of childhood CNS tumors and constitute a subtype of LGG that generally cannot undergo complete surgical resection. They can be bilateral, potentially arising anywhere along the optic pathway, although about 40–75% involve the chiasmatic-hypothalamic region [13,16,18,53]. This localization seems to be more aggressive and carries a worse prognosis, potentially involving the third ventricle and posterior optic tracts, and they usually present with the diencephalic syndrome (hypersomnia and cachexia) [18,53,54,55].

The treatment of OPG may be based on simply careful observation; however, patients presenting with severe clinical symptoms or mass effects with significant progressive growth require more aggressive treatment based on the use of chemotherapy, radiotherapy, and/or surgery [13,18,53,56].

OPGs can lead to variable clinical sequelae such as visual loss, endocrinopathies, and hypothalamic dysfunction, developmental/neuropsychological disorders, hydrocephalus, and focal neurological deficits because of their location [14,53,57].

Our patient suffering from OPG presented with rapidly progressing narcolepsy after several years of clinical and radiological stable disease. The narcolepsy diagnosis was confirmed by testing and had no improvement even though different lines of therapy were performed. Despite the relatively benign nature of the lesion, due to its dimension and hypothalamus involvement, it led to the death of our patient within five years from diagnosis. It is of interest to note that the patient did not have the typical suprasellar obesity.

## 4. Conclusions

Narcolepsy is a rare condition in pediatrics. It is mainly idiopathic, but we must not forget that it can result from conditions that damage the hypothalamic area such as CNS tumors. In a broad spectrum of further clinical perspectives, it may be important to include somnolence disturbances and narcolepsy as signs of tumor progression and using these clinical evidences at the same plan of ophthalmological/endocrinological impairments to evaluate an oncological treatment choice or change.

The treatment for narcolepsy is generally well tolerated with the resolution of symptoms; however, there are cases such as the case we have presented, resistant to treatment and with progressive and fatal evolution.

## Figures and Tables

**Figure 1 children-09-01455-f001:**
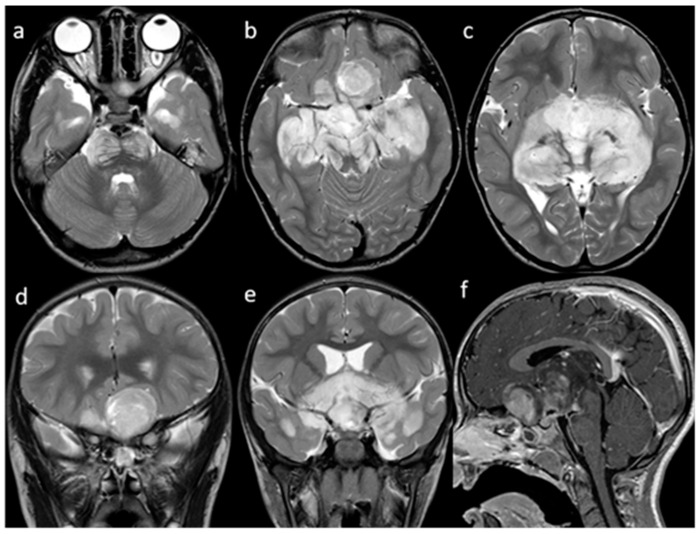
MRI. T2w axial (**a**–**c**) and coronal (**d**,**e**) images show a large, heterogeneously hyper-intense, optic-chiasmatic lobulated mass with massive involvement of the optic nerves and tracts, optic radiations, hypothalamus, temporal lobes, and brainstem. The mass compresses the frontal lobes and extends into the third ventricle, resulting in moderate enlargement of the lateral ventricles. Sagittal Gd T1w (**f**) image shows inhomogeneous contrast-enhancement of the suprasellar mass.

**Figure 2 children-09-01455-f002:**
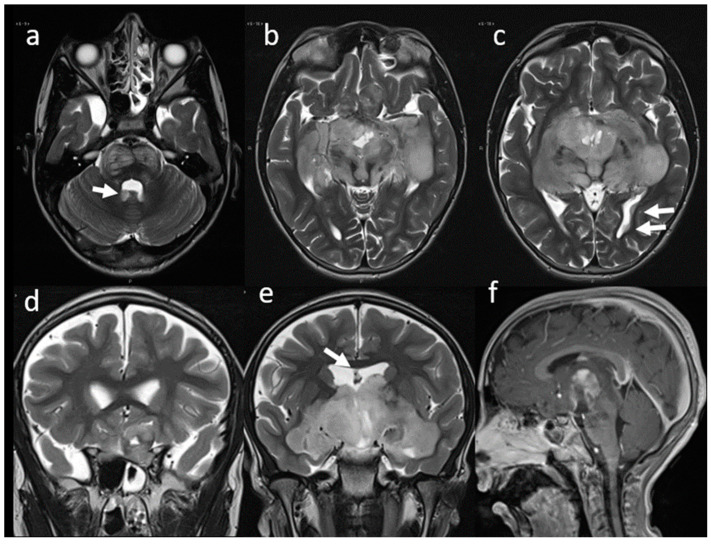
MRI. T2w axial (**a**–**c**) and coronal (**d**,**e**) images at the time of disease progression. Signal characteristics appear essentially unchanged. Multiple secondary localizations along the ventricular profiles are also well evident (arrows). Sagittal Gd T1w (**f**) image shows partial reduction of the inhomogeneous contrast enhancement of the suprasellar mass.

**Table 1 children-09-01455-t001:** Review of pediatric series of secondary narcolepsy.

Authors	N Patient	Median Age	Location of Tumor/Diagnosis	Sleep-Wake Characteristics
Pickering et al. (2021) [32]	61	12.4 years	66% sleep-wake regulatory areas (brain stem, basal forebrain, hypothalamus, thalamus); 34% other areas	90% of sleep disorders **8% narcolepsy**
Mandrell et al. (2020) [11]	110	10.3 years	Sellar/parasellar region	**27% narcolepsy** 35% hypersomnia
Weil et al. (2018) [30]	1	11 years	Sellar/parasellar region	**narcolepsy**
Rosen et al. (2011) [38]	70	10 years	68% CNS (hypothalamus/brainstem, posterior fossa, cortex) 32% no CNS (leukemia, upper airway, kidney)	**80% narcolepsy** in CNS location
Rosen et al. (2003) [35]	14	11 years	Sellar/parasellar region	**55% narcolepsy**

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
