# Peer review of "Secondary Narcolepsy as Worsening Sign in a Pediatric Case of Optic Pathway Glioma"

_children, 2022, doi:10.3390/children9101455_

Round 1

Reviewer 1 Report

This article by cacchione et al., is a case study of a 3-year-old boy with impairments and a history of seizures. MRI was performed that showed lesion involving optic chiasmatic region. Biopsy was performed that identified glioma. Chemotherapy was started following which the disease progressed after a year with cachexia was shown by brain MRI. Second line chemotherapy was started. At 8 years of age, the patient came to emergency, CT and MRI showed stable disease. MSLT showed narcolepsy pattern. Thus, the clinical manifestations, treatment and the effects observed are discussed in detail this paper. Other neurological symptoms are also documented. Unfortunately, these observations led to the demise of the child. Since narcolepsy is a rare occurrence with OPGs, this study would add up to literature of these disease conditions that are scarce.  In discussion, other studies are also tabulated.

Suggestions:

Include why auto antibody testing was performed.

Please read the full article and follow same tense (past tense) throughout the paper.

Also the paper requires a spell check.

Author Response

REVIEWER #1: " This article by cacchione et al., is a case study of a 3-year-old boy with impairments and a history of seizures. MRI was performed that showed lesion involving optic chiasmatic region. Biopsy was performed that identified glioma. Chemotherapy was started following which the disease progressed after a year with cachexia was shown by brain MRI. Second line chemotherapy was started. At 8 years of age, the patient came to emergency, CT and MRI showed stable disease. MSLT showed narcolepsy pattern. Thus, the clinical manifestations, treatment and the effects observed are discussed in detail this paper. Other neurological symptoms are also documented. Unfortunately, these observations led to the demise of the child. Since narcolepsy is a rare occurrence with OPGs, this study would add up to literature of these disease conditions that are scarce.  In discussion, other studies are also tabulated.

Suggestions:

Include why auto antibody testing was performed. Added in discussion line 141  Indirect evidence of an autoimmune pathogenesis of narcolepsy is provided by the observation that narcolepsy occasionally occurs in association with paraneoplastic syndromes and other autoimmune diseases, such as multiple sclerosis, coeliac disease and systemic lupus erythematosus. In our patient autoantibodies Anti GAD, Anti Hu, Anti Yu, Anti NMDA, and voltage-gated K channel antibody, exploring an autoimmune pathogenesis of somnolence resulted negative.

Please read the full article and follow same tense (past tense) throughout the paper. Also the paper requires a spell check. Reviewed by an english native language"

Reviewer 2 Report

The authors present an interesting manuscript regarding a rare complication in OPG; furthermore, they perform a literature review to complete their research. Nevertheless, the manuscript is not organized according CARE guidelines and this lack makes consultation less usable.

The case is well presented but some pitfalls can be detected and, therefore, ameliorated by the authors:

- Please quantify the sever bilateral hypovision (line 65); during case management and follow up, does this condition advance?

- Which surgical route was considered for the biopsy? Why do the authors affirm that biopsy confirmed ganglioglioma instead of diagnosed? Please provide molecular and histochemical information of this ganglioglioma and grading according to 2021 WHO CNS5 classification.

- I suggest the authors adding a further figure regarding the first MRI documenting tumor progression (line 82).

- How can the authors perform differential diagnosis between narcolepsy and comatose state due to hydrocephalus? This issue should be discussed in Case Presentation (lines 106-109).

In the Discussion the authors compare their case with literature series: first of all, they should add a column in Table 1 regarding the mortality of narcolepsy. Furthermore, beyond pharmacologic weapons (pay attention to the repetition at line 149), even surgical possibility should be discussed: does a cytoreductive surgery play a favorable role by means of releasing the hypothalamus from tumor compression? Was this option considered by the authors in this devastating tumor?

Finally, the draft lacks a synthesis of the literature review: by the way, which conclusions can be made after the review interpretation? How the literature series can integrate with this interesting case to provide relevant informations to the audience? In this sense, the Discussion should be rethought adding a paragraph regarding further clinical perspectives and considerations in case of this rare complication of a rare tumor entity.

Author Response

REVIEWER #2: 

"The authors present an interesting manuscript regarding a rare complication in OPG; furthermore, they perform a literature review to complete their research. Nevertheless, the manuscript is not organized according CARE guidelines and this lack makes consultation less usable.

The case is well presented but some pitfalls can be detected and, therefore, ameliorated by the authors:

- Please quantify the sever bilateral hypovision (line 65); added in line 72 (bilateral optic hypothrophy, optic coherence tomography highliting a retinal nerve fiber layer thickness (RNFL) severe reduction, visual loss quantifiable with Lea figures in 2/10 in left eye and 4/10 in right eye) horizontal nystagmus, divergent strabismus of the left eye and poor pupillary reaction to photostimulation). ; during case management and follow up, does this condition advance? Yes, at the moment of radiological progressione a complete hypovision was reached (line 94-95)

- Which surgical route was considered for the biopsy? ​Stereotactic robot-assisted needle biopsy (Rosa, Zimmer Biomet) was obtained by a right transfrontal route. Why do the authors affirm that biopsy confirmed ganglioglioma instead of diagnosed? changed the word with ALLOWED the diagnosis of...  Please provide molecular and histochemical information of this ganglioglioma ( added:  positivity for CD34+, GFAP  and OLIG2; p53 negative, BRAFV600  negative both in immunoistochemistry and molecular analysis, line 91) and grading according to 2021 WHO CNS5 classification.( grade 1 according the  latest edition of the World Health Organization (WHO) classification of central nervous system tumors)   

- I suggest the authors adding a further figure regarding the first MRI documenting tumor progression (line 82). Added other 3 figures documenting tumor progression (figure 2)

- How can the authors perform differential diagnosis between narcolepsy and comatose state due to hydrocephalus? This issue should be discussed in Case Presentation (lines 106-109). ​The possibility of hydrocephalus contributing to clinical deterioration was considered but MR did not show any signs of raised intracranial pressure.

In the Discussion the authors compare their case with literature series: first of all, they should add a column in Table 1 regarding the mortality of narcolepsy. No data on mortality secondary to narcolepsy in the pediatric age are avalilable. Mortality in adults is affected by metabolic syndrome and other problems associated with hypersomnia. Narcolepsy, istead, is clearly associated with significant morbidities. The largest pediatric serie has been analysed from the Danish National Patient Registry (NPR) with  243 patients diagnosed with narcolepsy. The patients showed elevated odds ratios of endocrine and metabolic conditions, nervous disorders,psychiatric illnesses, but not conclusion on the real impact of narcolepsy on mortality was available [Jennum P, Pickering L, Thorstensen EW, Ibsen R, Kjellberg J. Morbidity of childhood onset narcolepsy: a controlled national study. Sleep Med. 2017 Jan;29:13-17. doi: 10.1016/j.sleep.2016.09.013. Epub 2016 Nov 3. PMID: 28153208.]

    Furthermore, beyond pharmacologic weapons (pay attention to the repetition at line 149) corrected, even surgical possibility should be discussed: does a cytoreductive surgery play a favorable role by means of releasing the hypothalamus from tumor compression? Was this option considered by the authors in this devastating tumor? ​Surgical debulking of optic pathway glioma has been previously reported and has a role in the management of elevated intracranial pressure and compression symptoms. The possibility of a surgical procedure was discussed at the Tumor Board and considered futile since the tumor had an infiltrative growth and imaging did not show any signs of intracranial hypertension.

Finally, the draft lacks a synthesis of the literature review: by the way, which conclusions can be made after the review interpretation? How the literature series can integrate with this interesting case to provide relevant informations to the audience? In this sense, the Discussion should be rethought adding a paragraph regarding further clinical perspectives and considerations in case of this rare complication of a rare tumor entity." In conclusions added the sentence: in a broad spectrum of further clinical perspectives it may be important to include somnolence disturbances and narcolepsy as signs of tumor progression and using these clinical evidences at the same plan of opthalmological/endocrinological impairments to evaluate an  oncological treatment choice or change

Reviewer 3 Report

In this manuscript, the authors Beatrice Laus et. al, summarized a non-NF1 patient suffering from a very extensive optical pathway glioma who several years after diagnosis in radiological condition of stable diseases presented with severe narcolepsy, a rare complication, which led to the death of the patient. The phenomenon is very interesting and valuable in clinic process for the kids, especially who can’t speak. Furthermore, the authors analyzed the potential reasons of narcolepsy in this case. During this part, did the authors have the evidence about the electrolytes test, such as potassium ion ,sodium ion, or hemoglobin in the blood? Because lack of blood K+, Na+,  H2O, or anemia when the tumor is big enough, the kid are very easy to have sleepy syndrome. I have no other questions about this manuscript.

Author Response

Thank you very much for your comments.

No other systemic causes of narcolepsy were found in our patient.

According to your valuable suggestion, we added this sentence to the paper: Other systemic causes of somnolence such as electrolyte imbalance or anemia were excluded. 

Round 2

Reviewer 2 Report

The authors complete an accurate process of revision, according also to my suggestions, improving the overall quality of this manuscript; therefore, I think that the draft presents an archival value and should be considered for publication.

Author Response

Thank you very much for your comments.

We revised the language according to your suggestion.